# Preparation and Application of Bismuth/MXene Nano-Composite as Electrochemical Sensor for Heavy Metal Ions Detection

**DOI:** 10.3390/nano10050866

**Published:** 2020-04-30

**Authors:** Ying He, Li Ma, Liya Zhou, Guanhua Liu, Yanjun Jiang, Jing Gao

**Affiliations:** School of Chemical Engineering and Technology, Hebei University of Technology, 8 Guangrong Road, Hongqiao District, Tianjin 300130, China; heying1980@hebut.edu.cn (Y.H.); malihebut@163.com (L.M.); liyazhou@hebut.edu.cn (L.Z.); ghliu@hebut.edu.cn (G.L.)

**Keywords:** MXene, Ti_3_C_2_T*_x_*, BiNPs, electrochemistry, heavy metal ions

## Abstract

A nano-form composite of MXenes (Ti_3_C_2_T*_x_*, T*_x_* = -O, -OH, -F) was synthesized through depositing bismuth-nanoparticle (BiNPs) onto Ti_3_C_2_T*_x_* sheets. Because of the preventive effect of the two-dimensional layered structure of Ti_3_C_2_T*_x_*, the nanoparticles of Bi were uniform and well attached on the Ti_3_C_2_T*_x_*. The obtained BiNPs/Ti_3_C_2_T*_x_* nano-composite was applied for sensors construction of electrochemical detecting of Pb^2+^ and Cd^2+^ heavy metal ions. The produced BiNPs@Ti_3_C_2_T*_x_*-based sensor showed high effective surface area and excellent conductivity. Also, the BiNPs were efficient for anodic-stripping voltammetric to detect heavy metal ions. After conditions optimization, the BiNPs@Ti_3_C_2_T*_x_* nano-sensor could detect Pb^2+^ and Cd^2+^ simultaneously and the detection limits were 10.8 nM for Pb^2+^ and 12.4 nM for Cd^2+^. The BiNPs@Ti_3_C_2_T*_x_* was promising for detecting heavy metal ions due to their high surface area, fast electron-transfer ability, environmental friendliness, and facial preparation.

## 1. Introduction

Heavy metal ions and non-degradable ions, such as lead (Pb), cadmium (Cd), mercury (Hg), and chromium (Cr), do not decompose for decades or even centuries. This can produce various diseases for human health and other living environments [1]. Also, these ions in the environment is very dangerous and we need to remove it before accumulation. The development of sensitive approaches for detecting and determining toxic heavy metal ions is an urgent need [2,3,4]. Several analytical methods including Inductively Coupled Plasma-Mass Spectrometry (ICP-MS), Atomic Absorption Spectroscopy (AAS), X-ray Fluorescence Spectrometer (XRF), and Inductively Coupled Plasma Optical Emission Spectrometer (ICP-OES) are used for testing of these ions, but they do not have widespread applications due to their high cost and they require trained people to have contact with the equipment [5,6]. Thus, developing simple and cheap methods for sensitive detection of heavy metal ions is required. Electrochemical technique plays a critical role in detecting heavy metal ions due to its low cost, high sensitivity, rapid performance, and portability [7,8,9].

Recently, bismuth-based electrodes have become an alternative approach to mercury-based ones with some advantages like their environmental friendliness and excellent resolution of neighboring peaks. Because of having a highly specific surface and plentiful active sites, bismuth nanoparticles (BiNPs) have higher electro-analytical and catalytic activity than the traditional Bi films [10,11]. Until now, different BiNPs have been used to construct bismuth-based electrodes for electrochemical measurements. After modifying BiNPs, the behavior of the electrodes will be improved due to the newly forming multicomponent alloys and the combination of the nanomaterials properties [12]. However, due to a high surface free energy, BiNPs easily tend to aggregate and result in reduced original availability and decreased performance. A credible strategy to address this problem is anchoring BiNPs on specific supports like carbon nanomaterials such as graphene, carbon nanotube, graphite nanofiber, and Metal–organic frameworks (MOFs)-derived porous carbons [13].

As a kind of novel two-dimensional materials, MXenes have gained growing attention in supercapacitors, batteries, transparent electrodes, sensors, and catalysts. It has many positive properties such as excellent electrical conductivities, good structure, large hydrophilic surface area, and chemical stability [14]. In our previous report, acetylcholinesterase biosensor modified with Ag@MXene nano-composite was prepared for malathion detection and the obtained sensor exhibited good sensitivity at low levels and acceptable stability [15]. However, as far as we know, there are no reports regarding anchoring BiNPs onto Ti_3_C_2_T*_x_* MXene nano-sheets for constructing electrochemical sensors for detecting heavy metal ions. Thus, in this work, the BiNPs/Ti_3_C_2_T*_x_* nano-composite was designed, synthesized and applied for glassy carbon electrode (GCE) development as an electrochemical sensor for assaying of heavy metal ions. Also, the Pb^2+^ and Cd^2+^ ions in different water kinds were determined by the obtained electrochemical sensor and the sensitivity, selectivity and repeatability properties were studied.

## 2. Experimental Section

### 2.1. Materials

Analytical grade Pb(NO_3_)_2_, Bi(NO_3_)_3_, CdCl_2_, FeCl_3_, MgCl_2_, CoCl_2_, ZnCl_2_, MnCl_2_, CuSO_4_, Al(NO_3_)_3_, Ni(NO_3_)_2_, H_2_SO_4_, HNO_3_, C_6_H_5_Na_3_O_7_, NaBH_4_, CH_3_COOH, chitosan(CS), absolute ethanol, and polyethylene glycol were purchased from Fengchuan Chemical Reagent Co. Ltd., Tianjin, China. Acetate buffer (0.1 M) solution with different pH levels was prepared from stock solutions of 0.1 M sodium acetate and acetic acid and then applied as a supporting electrolyte agent.

#### 2.1.1. Synthesis of BiNPs@Ti_3_C_2_T*_x_* Nano-Composite

Ti_3_C_2_T*_x_* was prepared through selective etching of Al from Ti_3_AlC_2_ according to our previous report [16], and the detailed synthesis procedures were displayed on the Appendix A. The BiNPs@Ti_3_C_2_T*_x_* nano-composite was obtained by mixing Ti_3_C_2_T*_x_* nano-sheets with Bi as follows: 2 mg of Ti_3_C_2_T*_x_* was dispersed into 5 mL of ethylene glycol by ultra-sonication for 1 h, and then 5 mg of C_6_H_5_Na_3_O_7_ was dissolved into the dispersed solution; 5 mL of Bi(NO_3_)_3_·5H_2_O (16.3 mg) was added into the mixture. After stirring for 12 h, 0.5 mL of 0.1 M NaBH_4_ was added with shaking for 1 h. The solid products were precipitated using centrifugation, and then washed three times by distilled water and then absolute ethanol. The obtained nano-composite was dried in a vacuum oven at 50 °C for 24 h.

#### 2.1.2. Preparation of BiNPs@Ti_3_C_2_T*_x_*/GCE Sensor

For preparing sensors, GCE was polished with alumina slurries mechanically and then sequentially sonicated with absolute ethanol, 0.05 M HNO_3_ and distilled water separately and sequentially. The refined GCE was activated by H_2_SO_4_ (0.5 M). The BiNPs@Ti_3_C_2_T*_x_* nano-composite was added into 0.20% CS solution to obtain homogeneous suspension. A suspension aliquot was drop-cast on the refined GCE and dried in air. The obtained BiNPs@Ti_3_C_2_T*_x_*/GCE sensor was stored until used.

### 2.2. Electrochemical Measurements

Electrochemical performance of the produced nano-sensor was evaluated by a CHI660E electrochemical workstation (Shanghai Chenhua, China). The electrochemical measurements were conducted on a three electrode system; BiNPs@Ti_3_C_2_T*_x_*/GCE (prepared electrode), platinum wire (counter electrode) and saturated calomel electrode (SCE, reference electrode). The tested electrochemical techniques were cyclic voltammograms (CV), electrochemical impedance spectra (EIS) and chronocoulometry (CC). CV was carried out at a scan rate of 50 mV s^−1^, and EIS was performed with afrequency that changed from 10^−2^ to 10^5^ Hz with a signal amplitude of 5 mV. The electrochemical properties of the produced sensor were determined according to previous studies [11].

The active area of BiNPs@Ti_3_C_2_T*_x_* nano-composite was calculated using the following equation:Q(t)=2nFAcD12t12π12+Qdl+Qads
where *Q* is the absolute value of the reduction charge; *n* is the number of electrons transfer; *F* is the Faraday constant (96,485 C/mol); *A* is the effective area; *c* is the substrate concentration; *D* is the diffusion coefficient (7.6 × 10^−6^ cm^2^/s); *t* is the time; *Q_dl_* is the double-layer charge; and *Q_ads_* is the Faradaic charge consumed by adsorbed species.

### 2.3. Characterization

The morphology of Ti_3_C_2_T*_x_* nano-sheets and BiNPs@Ti_3_C_2_T*_x_* nano-composite was characterized using Scanning Electron Microscope (SEM) (NanoSEM45011, FEI, Portland, OR, USA) and Transmission Electron Microscope (TEM) (JEM-2100, JEOL Inc., Tokyo, Japan). Elemental compositions of the samples were analyzed by Energy Dispersive X-Ray Spectroscopy (EDX) spectroscopy (EDAX Inc., Mahwah, NJ, USA). The crystallinity of Ti_3_C_2_T*_x_* nano-sheets and BiNPs@Ti_3_C_2_T*_x_* nano-composites was determined by XRD diffractometer (D8 Advance, Bruker, karlsruhe, Germany). The chemical state of the samples was demonstrated using XPS (Da Vinci, Bruker, karlsruhe, Germany).

### 2.4. Application of BiNPs@Ti_3_C_2_T_x_-Based Sensorfor Heavy Metal Ions Detection

To investigate the applicability of the prepared sensor, two kinds of water samples (tap and lake water) were used. Standard solutions of Pb^2^^+^ and Cd^2^^+^ were added. BiNPs@Ti_3_C_2_T*_x_*-based sensor determined Pb^2+^ and Cd^2+^ ions through the prestigious square wave anodic stripping voltammetry (SWASV) technique [2]. SWASV was applied at a deposition potential of −1.0 V for 300 s in 0.1 M NaAc/HAc buffer (pH 5.0), and performed in the potential range of −0.95 to −0.35 V with a frequency of 15 Hz, an amplitude of 25 mV, and an increment potential of 4 mV. For comparison, ICP-MS standard method was used to confirm the concentrations of heavy metal ions found in tap and lake waters.

## 3. Results and Discussion

### 3.1. Preparation and Characterization of BiNPs@Ti_3_C_2_T_x_

Heavy metal ions are environmental dangerous ions due to their non-degradability properties, causing harmful diseases for human and not decomposability. For that, researchers need to detect these metals by low levels in the environment; the suitable technology for that is sensors. In the current work, a nano-composite-based sensor was designed and produced. Ti_3_C_2_T*_x_* nano-sheets were prepared using selective etching method [16] and then characterized by SEM, TEM and EDX instruments. As shown in Figure 1A, the prepared Ti_3_C_2_T*_x_* nano-sheets displayed typical two-dimensional layered structures. It appears as homogeneous, continuous, obtains a large area, and facilitates the mass transfer [17]. When Bi(NO_3_)_3_ was added to the reaction, Bi^3+^ ions could be adsorbed and attached on Ti_3_C_2_T*_x_* nano-sheets through electrostatic interaction [18] and then the Ti_3_C_2_T*_x_* nano-sheets served the nucleation sites of BiNPs. Because of the reduction action by NaBH_4_, BiNPs was grown on the surface of Ti_3_C_2_T*_x_* nano-sheets [13]. Figure 1B indicated that ununiform Bi nanoparticles were dispersed on the surface or interlayers of Ti_3_C_2_T*_x_* nano-sheets. To further investigate the microstructure of BiNPs@Ti_3_C_2_T*_x_*, TEM examination was conducted and the resulted picture indicated that the BiNPs (dark dots) were distributed on the Ti_3_C_2_T*_x_* nano-sheets (Figure 1C) and un-agglomerated nanoparticles could be observed. In the case of elemental analyses, EDX mapping revealed that the BiNPs@Ti_3_C_2_T*_x_* nano-composite involved Bi and confirmed the attaching of BiNPs onto Ti_3_C_2_T*_x_* nano-sheets (Figure 1D). In addition, the layered morphology of Ti_3_C_2_T*_x_* displayed a few changes after the formation of Bi nanoparticles. This may be due to the attachment of Bi on the surface of Ti_3_C_2_T*_x_* nano-sheets and making some changes in the sheets surface [19]. The elemental analysis confirmed that the elements Ti, Al, Si, C, F, O, and Bi were detected by deferent percentages.

For the crystal structure of Ti_3_C_2_T*_x_* and BiNPs@Ti_3_C_2_T*_x_*, XRD pattern of pristine Ti_3_C_2_T*_x_* nano-sheets as presented in Figure 2A(b) showed three peaks of projecting diffraction corresponding to (002), (004) and (110) intensity, respectively. Meanwhile, the XRD pattern of BiNPs@Ti_3_C_2_T*_x_* (Figure 2A(a)) showed obvious peaks at 27.2, 38.1 and 39.7°, which were assigned to (012), (104), and (110) planes of the rhombohedra crystal structure of Bi. The successful deposition of BiNPs on Ti_3_C_2_T*_x_* is demonstrated. For XPS spectrum (Figure 2B), it is shown that both Ti_3_C_2_T*_x_* nano-sheets and BiNPs@Ti_3_C_2_T*_x_* were composed of Ti, C, O, and F. It is in agreement with previous reports [20]. Moreover, the XPS spectra of BiNPs@Ti_3_C_2_T*_x_* showed the presence of Bi, suggesting that BiNPs@Ti_3_C_2_T*_x_* had been synthesized successfully. As shown in Figure 2C, the peaks of Bi 4f (4f_7/2_ and 4f_5/2_) were qualified to Bi (0), which indicated that nearly all bismuth in the BiNPs@Ti_3_C_2_T*_x_* was Bi (0). In brief, all these results confirmed the successful preparation of BiNPs@Ti_3_C_2_T*_x_*.

### 3.2. Electrochemical Characterization of the BiNPs@Ti_3_C_2_T_x_/GCE

The basic electrochemical techniques like cyclic voltammograms (CV), electrochemical impedance spectra (EIS) and chronocoulometry (CC) were conducted to study the properties of BiNPs@Ti_3_C_2_T*_x_*/GCE sensor. CV and EIS were realized in a 5 mM [Fe(CN)_6_]^3−/4−^ with 0.1 M KCl. Three peak currents of increased oxidation are shown in Figure 3A, which reached 53.6 μA (GCE bare), 130.5 μA (Ti_3_C_2_T*_x_*/GCE) and 162.9 μA (BiNPs@Ti_3_C_2_T*_x_*/GCE). These results showed that BiNPs@Ti_3_C_2_T*_x_* exhibited an excellent electrical conductivity and influenced a large area. In addition, EIS was used for studying the interfacial electron transfer resistance (*R*_et_). The data in Figure 3B showed three *R*_et_ of 1500 U on GCE bare, 720 U on Ti_3_C_2_T*_x_*/GCE and 450 U on BiNPs@Ti_3_C_2_T*_x_*/GCE, respectively, were obtained. The reduced R_et_ was attributed to the distended high electrical conductivity and specific area, which can improve the transfer of electron and mass exchange of electro-active indicators on the surface [21]. BiNPs@Ti_3_C_2_T*_x_* nano-composite’s effective area was measured by CC tests in 0.1 mM K_3_Fe(CN)_6_ solution, and the result is illustrated in Figure 3C.

As illustrated in Figure 3D, the effective area of BiNPs@Ti_3_C_2_T*_x_*/GCE reached 0.04857 cm^2^, which is bigger than that of GCE bare (0.01677 cm^2^) and Ti_3_C_2_T*_x_*/GCE (0.02681 cm^2^). All the above results demonstrated that the excellent properties including excellent electrical conductivity, high structural stability and large active area of Ti_3_C_2_T*_x_* and BiNPs were integrated into the BiNPs@Ti_3_C_2_T*_x_* nano-composite. This is closely in agreement with the results obtained by L. Shi, et al. [13].

### 3.3. Optimization of Experimental Conditions

Benefitting from the high active area of BiNPs@Ti_3_C_2_T*_x_*/GCE, highly active electrochemical sensors were built for the simultaneous detection of Pb^2+^ and Cd^2+^ with SWASV. As presented in Figure 4, different electrodes were used to detect Pb^2+^ and Cd^2+^. Comparing with the responses of GCE bare and Ti_3_C_2_T*_x_*/GCE, the BiNPs@Ti_3_C_2_T*_x_*/GCE composite showed well-defined two individual voltammetric peaks at −0.57 V (Pb^2+^) and −0.78 V (Cd^2+^) with higher signals, representative of better detection choosiness and exactness of BiNPs@Ti_3_C_2_T*_x_*/GCE toward the Pb^2+^ and Cd^2+^. This phenomenon could be attributed to the integration and synergic effect of Ti_3_C_2_T*_x_* and BiNPs, which was a benefit of the higher ion adsorption, resulting in a higher response of Pb^2+^ and Cd^2+^ detection [22].

The pH level is a crucial factor for heavy metal ions analysis, as well as the behavior of electrochemical sensor responding. The effect of pH levels (ranging from 3.0 to 6.0) on responses to the obtained electrochemical sensor was investigated in the present work. Figure 5A shows that increasing signals when pH increased from 3.0 to 5.0, and decreased sharply when pH changed from 5.0 to 6.0. This can be explained as follows: at lower pH values, the protonation on the BiNPs@Ti_3_C_2_T*_x_*/GCE composite resulted in weak signals; when the pH was higher than 5.0, the hydrolysis occurred and then induced a decrease of signals [23]. In the case of the deposition potential, the effect of deposition potential on response of signals was investigated and the stripping peak currents exhibited increased value obviously at −1.0 V, while deposition potentials were more negative than −1.0 V (Figure 5B). The good hydrogen evolution was enhanced, leading to the failure of signals response [24]. Thus, −1.0 V was selected to be the optimum admission potential. Moreover, as shown in Figure 5C, the enlarging of signals response from 100 to 500 s was observed. The time of deposition (300 s) was chosen for the following experiments due to the fact that a higher response signal under shorter deposition time was expected in applications.

### 3.4. Analytical Performance of the Produced Electrochemical Sensor

As shown in Figure 6, the sensor’s sensitivity and linear range for Pb^2+^ and Cd^2+^ were evaluated in the current work. With the increase of ion concentrations, the response of signals was enhanced. Both Pb^2+^ and Cd^2+^ stripping peak currents were increased proportionally with the increasing concentrations of Pb^2+^ and Cd^2+^ ions. The calibration curves of Pb^2+^ and Cd^2+^ were linear from 0.06 to 0.6 μM and 0.08 to 0.8 μM, respectively. The linear equations were evaluated as $\frac{I}{\mu A}=1.531+42.85C/\mu M$ and $\frac{I}{\mu A}=1.895+35.08C/\mu M$ for Pb^2+^ and Cd^2+^, with correlation coefficients of 0.9995 and 0.9976, respectively. The detection limits were recorded to be 10.8 nm for Pb^2+^ and 12.4 nm for Cd^2+^, and it is much lower than the recommended concentrations in drinking water as noted by the World Health Organization [25].

The joint interfering between Cd^2+^ and Pb^2+^ on BiNPs@Ti_3_C_2_T*_x_*/GCE was explored to evaluate the problem of mutual interference between heavy metal ions in immediate detection. The concentration of Cd^2+^ was fixed, while the concentration of Pb^2+^ was increased (Figure 7A). Also, the peak current of Pb^2+^ increased while the concentration of Cd^2+^ remained the same. However, the data in Figure 7B represented that when the concentration of Pb^2+^ is fixed, and increasing the concentration of Cd^2+^, the peak current of Pb^2+^ fluctuated narrowly while the peak current of Cd^2+^ increased. The calibration plots of the peak present against concentration exhibited good linearity with the ranges of 0.06 to 0.6 μM for Pb^2+^ and 0.08 to 0.6 μM for Cd^2+^. The linear equations are $\frac{I}{\mu A}=-0.4056+\frac{39.64C}{\mu M}$ and $\frac{I}{\mu A}=-1.584+\frac{34.39C}{\mu M}$ for Pb^2+^ and Cd^2+^, with correlation coefficients of 0.9951 and 0.9992, respectively. The detection slopes of both Pb^2+^ and Cd^2+^ were similar to data illustrated in Figure 6B (39.64 versus 42.85 for Pb^2+^ and 34.39 versus 35.08 for Cd^2+^), and thus, designated that no joint interference between Pb^2+^ and Cd^2+^ occurred at BiNPs@Ti_3_C_2_T*_x_*/GCE.

### 3.5. Selectivity, Reusability and Stability of the Produced Sensor

Selectivity is the main factor in the detection of heavy metal ions with the SWASV technique. Samples of Mg^2+^, Fe^3+^, Co^2+^, Ni^2+^, Mn^2+^, Zn^2+^, Al^3+^, and Cu^2+^ ions with a concentration of 10 μM were added to explore the ions interference in an immediate detection of Pb^2+^ and Cd^2+^. The result in Figure 8 shows little changes in response signals except for Cu^2+^. However, in the state of adding Cu^2+^, the signals were significantly reduced because of the formation of intermetallic compounds. This phenomenon could be avoided through adding ferrocyanide to exclude the interference of Cu^2+^.

In order to ensure the reusability of the prepared sensor, it was determined at a concentration of 0.2 μM Pb^2+^ and 0.4 μM Cd^2+^, and the acceptable relative standard deviations (RSDs) were achieved with 3.7% Pb^2+^ and 2.7% Cd^2+^. Fortunately, after ensuring the reproducibility of the obtained sensor, similar responses were obtained by each electrode and the RSDs of 4.0% Pb^2+^ and 4.6% Cd^2+^. These results indicated that the obtained sensor has excellent reproducibility characteristics. Furthermore, the stability of sensor was examined for storing it in the refrigerator in dry state (4 °C). After 6 weeks of storing, 95.2% and 94.5% of the initial response sensitivity can be retained for Pb^2+^ and Cd^2+^, respectively. The high stability character was ascribed to the structure stability nature and binding strength of BiNPs@Ti_3_C_2_T*_x_* nano-composite on GCE [26].

### 3.6. Application of BiNPs@Ti_3_C_2_T_x_ Sensor forHeavy Metal Ions Detection in Tap and Lake Water

Heavy metal ions such as Pb^2+^ and Cd^2+^ are considered more dangerous for humans and other living organisms, because it causes many diseases. For this reason, it should be removed from all environments. In this work, BiNPs@Ti_3_C_2_T*_x_* sensor was produced for heavy metal ions detection at the lowest levels. Two heavy metal ions, Pb^2+^ and Cd^2+^, were tested for detection in two deferent water types (tap and lake water). The concentrations of Cd^2+^ and Pb^2+^ were 150 nM. As illustrated in Table 1, the recoveries of the detected heavy metalions were reached at 98.3% and 101.2% for Cd^2+^ and Pb^2+^, respectively in tap water, while it was 101.5% and 106.3% for Cd^2+^ and Pb^2+^, respectively, in lake water. In addition, the results of ICP-MS and the electrochemical analysis are in reasonably good agreement. These results confirmed the reliability and efficiency of the produced sensors for detection of Pb^2+^ and Cd^2+^ ions.

In another case, the analytical performance of the obtained electrochemical sensor for Pb^2+^ and Cd^2+^ detection is presented in Table 2. The limits of detection (LOD) of the sensor described in this work are lower than some of the previous works. Thus, the optimized BiNPs@Ti_3_C_2_T*_x_* showed acceptable results. Considering the acceptable LOD and the high recovery values, the developed sensor has a great potential for detecting Pb^2+^ and Cd^2+^ in practical water samples.

## 4. Conclusions

The BiNPs@Ti_3_C_2_T*_x_* nano-composite that integrated the advantages of BiNPs and Ti_3_C_2_T*_x_* with high surface area and good conductivity was prepared. BiNPs@Ti_3_C_2_T*_x_* was used to modify the electrochemical electrode for heavy metal ions detection as sensing technology. The obtained sensor can simultaneously detect Pb^2+^ and Cd^2+^ with high sensitivity and good exactness. However, Cu^2+^ can cause interference in this detection process. Considering the high effective surface area and excellent conductivity of the BiNPs@Ti_3_C_2_T*_x_*, the nano-composite can be applied as an excellent electrode-modification material for fast and convenient determination of heavy metal ions in environmental systems.

## Figures and Tables

**Figure 1 nanomaterials-10-00866-f001:**
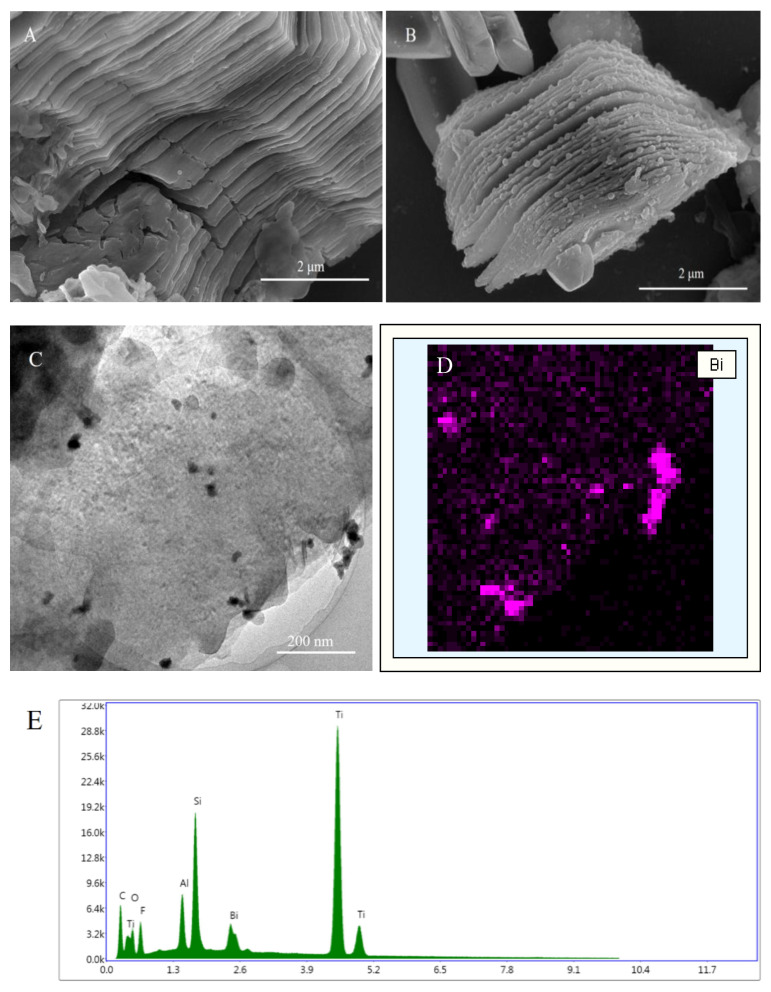
SEM images of Ti_3_C_2_T*_x_* (**A**) and BiNPs@Ti_3_C_2_T*_x_* (**B**); TEM image of BiNPs@Ti_3_C_2_T*_x_* (**C**); EDX mapping of Bi (**D**) and EDX elemental mappings (**E**).

**Figure 2 nanomaterials-10-00866-f002:**
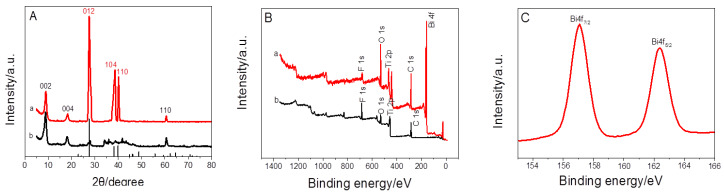
(**A**) XRD patterns of Ti_3_C_2_T*_x_* (b) and BiNPs@Ti_3_C_2_T*_x_* (a), (**B**) XPS spectra of Ti_3_C_2_T*_x_* (b) and BiNPs@Ti_3_C_2_T*_x_* (a), (**C**) Bi4f XPS spectrum of Bi.

**Figure 3 nanomaterials-10-00866-f003:**
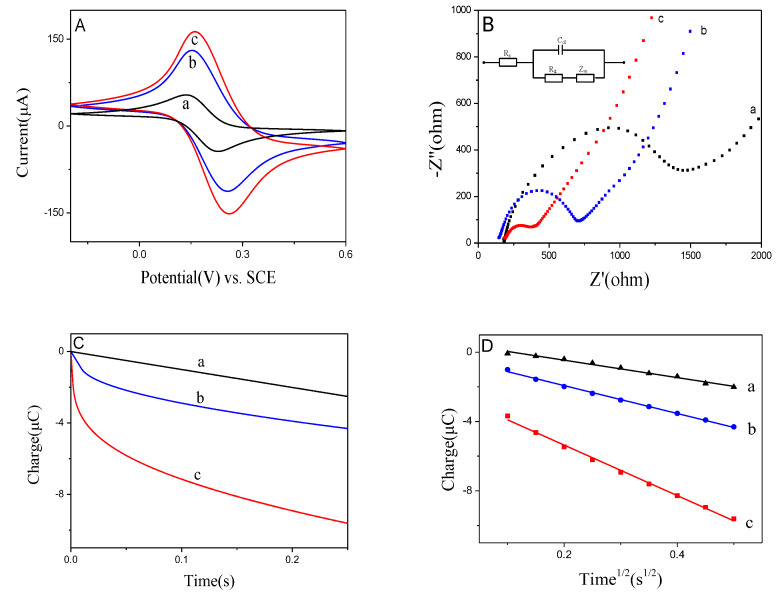
(**A**) Cyclic Voltammograms of GCE bare (a), Ti_3_C_2_T*_x_*/GCE (b) and BiNPs@Ti_3_C_2_T*_x_*/GCE (c)in 0.1M KCl containing 5 mM Fe(CN)_6_^3−/4−^; (**B**) EIS spectra of GCE bare (a), Ti_3_C_2_T*_x_*/GCE (b) and BiNPs@Ti_3_C_2_T*_x_*/GCE (c)in 0.1M KCl containing 5 mM Fe(CN)_6_^3−/4−^; (**C**,**D**) Plots of Q-t and Q-t^1/2^ of the tested electrodesin 0.1 M KCl containing 0.1 mM K_3_Fe(CN)_6_.

**Figure 4 nanomaterials-10-00866-f004:**
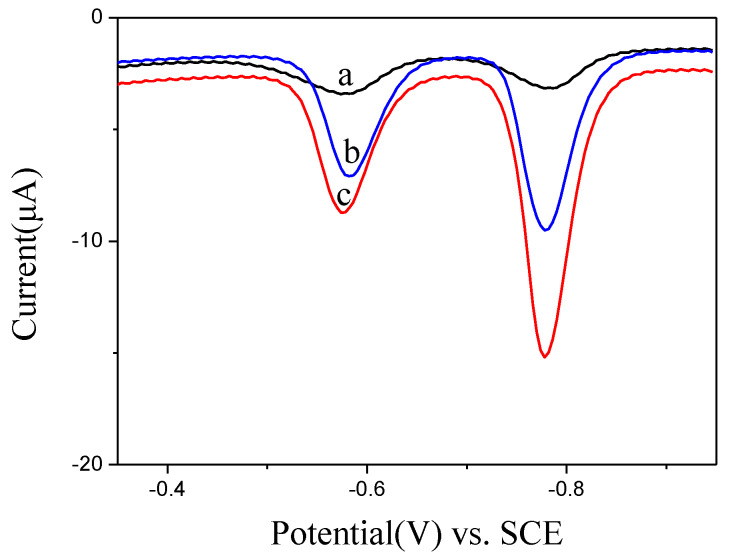
SWASV curves of 0.2 μM Pb^2+^ and 0.4 μM Cd^2+^ at GCE bare (a), Ti_3_C_2_T*_x_*/GCE (b) and BiNPs@Ti_3_C_2_T*_x_*/GCE (c) in 0.1 M acetate buffer solution (pH 5.0).

**Figure 5 nanomaterials-10-00866-f005:**
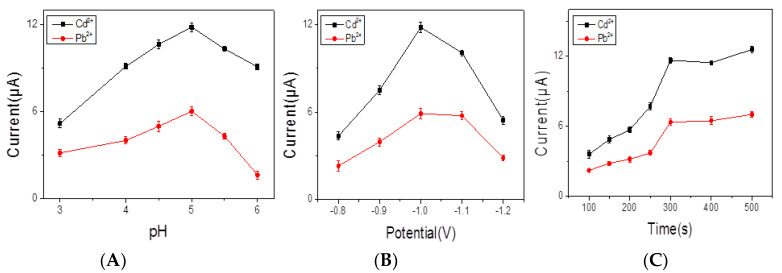
Effection of pH levels (**A**), deposition potentials (**B**), and deposition time (**C**) on SWASV response of 0.2 μM Pb^2+^ and 0.4 μM Cd^2+^ in acetate buffer at BiNPs@Ti_3_C_2_T*_x_*/GCE.

**Figure 6 nanomaterials-10-00866-f006:**
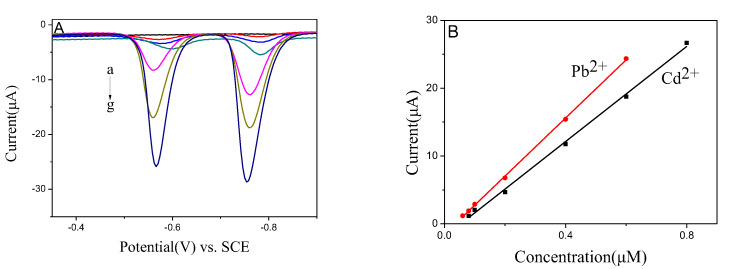
(**A**) SWASV responses of BiNPs@Ti_3_C_2_T*_x_*/GCE for the simultaneous analysis of Pb^2+^ and Cd^2+^ at 0, 0.06, 0.08, 0.1, 0.2, 0.4, and 0.6 μM Pb^2+^; and 0, 0.08, 0.1, 0.2, 0.4, 0.6, and 0.8 μM of Cd^2+^, respectively; (**B**) Calibration curves of Pb^2+^ and Cd^2+^.

**Figure 7 nanomaterials-10-00866-f007:**
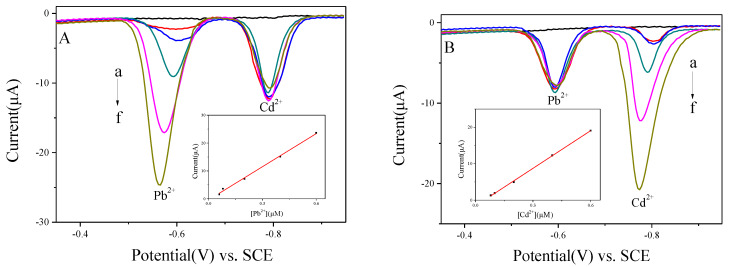
(**A**) SWASV curves of Pb^2+^ at 0, 0.06, 0.08, 0.2, 0.4, and 0.6 μM in the presence of 0.4 μM Cd^2+^ at BiNPs@Ti_3_C_2_T*_x_*/GCE, and the corresponding linear calibration plots against Pb^2+^. (**B**) SWASV curves of Cd^2+^ at 0, 0.08, 0.1, 0.2, 0.4, and 0.6 μM in the presence of 0.2 μM Pb^2+^ at BiNPs@Ti_3_C_2_T*_x_*/GCE, and the corresponding linear calibration plots against Cd^2+^.

**Figure 8 nanomaterials-10-00866-f008:**
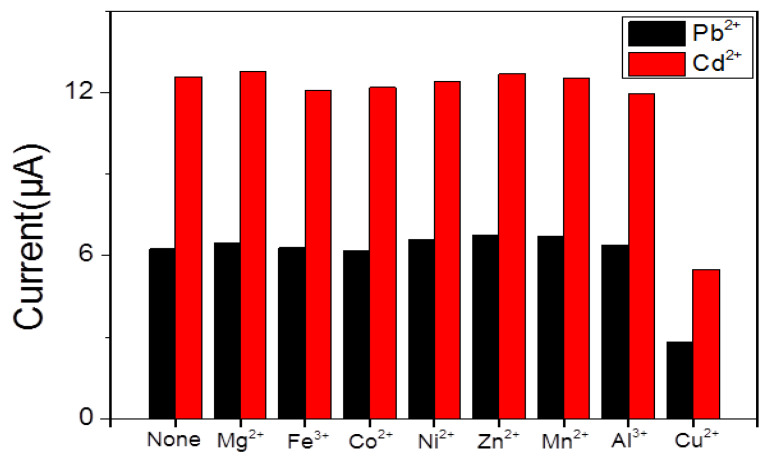
Selectivity of BiNPs@Ti_3_C_2_T*_x_*/GCE for simultanoues detection of Pb^2+^ and Cd^2+^.

**Table 1 nanomaterials-10-00866-t001:** Determination of Pb^2+^ and Cd^2+^ in water samples with the BiNPs@Ti_3_C_2_T*_x_*-based electrode.

Samples		Added (nM)	ICP-MS Result (nM)	Result (nM)	Recovery (%)
Tap water	Pb^2+^	150	149.6	151.8 ± 0.85	101.2
	Cd^2+^	150	150.8	147.4 ± 1.12	98.3
Lake water	Pb^2+^	150	151.6	159.4 ± 0.57	106.3
	Cd^2+^	150	151.2	152.2 ± 0.84	101.5

**Table 2 nanomaterials-10-00866-t002:** List of various electrochemical sensors for Pb^2+^ and Cd^2+^ detection.

Electrochemical Platform	Technique	Analyte (HMI)	LOD (ppb)
**Bi-C nano-composite [27]**	SWASV	Pb^2+^	0.65
		Cd^2+^	0.81
**RGO/Bi nano-composite [28]**	ASV	Pb^2+^	0.55
		Cd^2+^	2.80
**BiNPs [29]**	SWASV	Pb^2+^	2.00
		Cd^2+^	5.00
**Nano-Bi [30]**	SWASV	Pb^2+^	1.97
		Cd^2+^	2.54
**NPCGS/Bi nano-composite [11]**	SWASV	Pb^2+^	0.66
		Cd^2+^	0.46
**Ti_3_C_2_T*_x_*/BiNPs nano-composite**	SWASV	Pb^2+^	2.24
		Cd^2+^	1.39

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
