# Peer review of "Preparation and Application of Bismuth/MXene Nano-Composite as Electrochemical Sensor for Heavy Metal Ions Detection"

_nanomaterials, 2020, doi:10.3390/nano10050866_

Round 1
Reviewer 1 Report
The manuscript by Jing Gao et al. describes the preparation and study of BiNPs onto Ti3C2Tx MXene nano-sheets for the preparation of electrochemical sensors for the detection of heavy metal ions. The topic is interesting and it is well written but my main concern is related to certain aspects throughout the text, that have to be better explained and that I expose below. The conclusions, in my opinion, do not give an overview of the work.
In my opinion, the presented work is suitable for publication in Nanomaterials after to take into account the corrections that I indicate below.
Comments:
-Title: change “heavy metals” by “heavy metal ions”.
-Abstract: Ti3C2Tx, authors have to define the meaning of Tx.
-Experimental part:
Line 68, change “Ti3C2Txnano-composite” by “Ti3C2Tx nano-composite”.
Change” two by 2 and five by 5”
In my opinion is better use “Preparation” that “Fabrication”.
Result and discussion:
-Figure x should appear along the text as Figure x (without bold font).
-Pag 3, line 113, a reference should be added after “….method”. Line 118, Figure 2B should be Figure 1B.
-Pag 4, line 137, change “previous reports of [20].” by “previous reports [20]”.
-Pag 6, The authors assert, “the BiNPs@Ti3C2Tx/GCE composite showed well-defined two individual voltammetric peaks at 0.57 V (Pb2+) and 0.78 V (Cd2+) with higher signals, representative better detection choosiness and exactness of BiNPs@Ti3C2Tx/GCE toward the Pb2+ and Cd2+.” This is good, however the potential values appear as negative values in the figure 4, why?. Could the authors specify, next to the potential the corresponding redox couple?
-Pag 7. The sentence “Further increases in pH level, the hydrolysis was happened…” should be better explained.
Pag 9. In my opinion Cu2+ is an interference in the detection of Pb2+ and Cd2+. This aspect should be indicated in the conclusions.
-The authors have to define the meaning of RSDs.
-The authors assert: “After 6 weeks of storing, the sensor displayed an excellent stability with the RSDs of 1.76 % Pb2+ and 1.12 % Cd2+”, however these values have decreased from 3,7% to 1,76% and from 2,7% to 1.12%. Then, it can be said that this corresponds to excellent selectivity? an explanation has to be given.
-The authors write:” the analytical performance of the obtained electrochemical sensor for Pb2+ and Cd2+ detection was presented in the Table (2). The optimized BiNPs@Ti3C2Tx showed good limits of detection (LOD), because it is lower than the previous works in this field.
In my opinion this table should be discussed and the sentence should be changed since the LOD of the sensors described in this work are not lower than the previous works, only in some cases.
Reviewer 2 Report
The paper submitted by Y. Jiang and J. Gao describes a nanocomposite containing Bi nanoparticles deposited on Ti3C2Tx MXene and further used as sensor for simoultaneous detection of Pb2+ and Cd2+. The subject is of high interest taking into account the toxicity of those two metals and their wide distribution into environment.
In the present form this manuscript is not publishable. The English should be revised in order to be acceptable for publication. There are also other issues that should be addressed by the authors:
- In the Experimental part, please add the experimental conditions for electrochemical investigations (electrolytes, potentials, scan rate, frequency domain for EIS etc) for all the electrochemical techniques used to characterized and test the sensors.
- Fig 3. Legend- the Randles circuit for EIS should be added and more details about the experimental conditions in fig 3 legends.
- Chapter 3.3. - what reference electrode was used? In which electrolyte was performed the analysis? The pH?
- In fig. 3, 4, 6, 7 please add the reference when writing potential (for instance ....Potential (V) vs. Ag/AgCl?)
- It is usually calculated the detection limit not the discovery limit.
- The electrodes were kept in refrigerator in dry state?
- In table 1 are presented the results on real samples. Do the authors use other method to confirm the founded concentrations of heavy metals found in tap and lake waters?
Round 2
Reviewer 2 Report
The quality of the paper was improved but I have an observation, please pay attention to figures 3,4 and 6 where the old and new figures are mixed. Please erase the old figures.
